# Mesenchymal Stromal Cells Isolated from Ectopic but Not Eutopic Endometrium Display Pronounced Immunomodulatory Activity In Vitro

**DOI:** 10.3390/biomedicines9101286

**Published:** 2021-09-22

**Authors:** Alexey Yu. Lupatov, Roza Yu. Saryglar, Valentina V. Vtorushina, Rimma A. Poltavtseva, Oxana A. Bystrykh, Vladimir D. Chuprynin, Lyubov V. Krechetova, Stanislav V. Pavlovich, Konstantin N. Yarygin, Gennady T. Sukhikh

**Affiliations:** 1Institute of Biomedical Chemistry, Pogodinskaya 10, 119121 Moscow, Russia; roza_saryglar@mail.ru (R.Y.S.); kyarygin@yandex.ru (K.N.Y.); 2Research Center of Obstetrics, Gynecology, and Perinatology, Akademika Oparina 4, 117997 Moscow, Russia; v_vtorushina@oparina4.ru (V.V.V.); rimpol@mail.ru (R.A.P.); ksana.77@inbox.ru (O.A.B.); v_chuprynin@oparina4.ru (V.D.C.); l_krechetova@oparina4.ru (L.V.K.); st.pavlovich@mail.ru (S.V.P.); g_sukhikh@oparina4.ru (G.T.S.)

**Keywords:** endometriosis, mesenchymal stromal cells, dendritic cells, lymphocytes, immunosuppression, inflammation, secretome

## Abstract

A comparative analysis of the cell surface markers and immunological properties of cell cultures originating from normal endometrium and endometrioid heterotopias of women with extragenital endometriosis was carried out. Both types of cell cultures expressed surface molecules typical of mesenchymal stromal cells and did not express hematopoietic and epithelial markers. Despite similar phenotype, the mesenchymal stromal cells derived from the two sources had different immunomodulation capacities: the cells of endometrioid heterotopias but not eutopic endometrium could suppress dendritic cell differentiation from monocytes as well as lymphocyte proliferation in allogeneic co-cultures. A comparative multiplex analysis of the secretomes revealed a significant increase in the secretion of pro-inflammatory mediators, including IL6, IFN-γ, and several chemokines associated with inflammation by the stromal cells of ectopic lesions. The results demonstrate that the stromal cells of endometrioid heterotopias display enhanced pro-inflammatory and immunosuppressive activities, which most likely impact the pathogenesis and progression of the disease.

## 1. Introduction

Endometriosis is an estrogen-dependent disease characterized by a benign growth of abnormal endometrial tissue beyond the uterine cavity [1]. It is quite common among women of reproductive age. The estimated prevalence of endometriosis in the female population is about 1–10% depending on sampling and diagnostic methods [2,3]. Extragenital endometriosis usually occurs in the pelvic organs, but like cancer, it can spread seeding into the brain [4], lungs [5], or other sites. Circulating endometrial cells capable of disseminating were found in about 80% of patients with endometriosis [6]. Although the lethal outcomes in endometriosis are not typical, there are registered cases of malignant transformation of endometrioid heterotopias resulting in the formation of adenocarcinomas, or even sarcomas, arising from the stromal component of endometriotic lesions [7,8]. The most clinically relevant symptoms of endometriosis seriously affecting the quality of life and the incidence of depression among women are infertility and chronic pain associated with inflammation [9,10]. The conventional treatment of endometriosis includes hormonal therapy and surgery, but often it fails to prevent the recurrence of ectopic lesions. Further advanced research aimed at new insights into the pathogenesis of endometriosis is required to develop more effective treatments.

The mechanism of endometriosis progression is still not fully understood. The most widely accepted hypothesis suggests reflux of eutopic endometrial fragments through the fallopian tubes during menstruation and subsequent implantation onto the peritoneum and the ovary [11]. However, eutopic origin does not explain the incidence of endometriosis in newborns, prepubertal and postmenopausal females, women with congenital absence of the uterus, and even in men [12]. In these cases, endometriosis can be explained by its occurrence and development in situ. Within this concept, metaplasia of the coelomic epithelium or embryonic cell rests is considered the source of endometrioid cells [13]. It is possible that various pathogenetic mechanisms lead to the initiation of endometrioid heterotopias, where the accumulation of genetic alterations in the ectopic endometrium, including cancer-related mutations, causes their formation and progression. This assumption is supported by a significantly higher frequency of mutations in the endometrioid lesions than in the eutopic endometrium [14]. On the other hand, there is evidence that specific alterations in the eutopic endometrium are required to form heterotopias [15].

The endometrium contains stem cells, which can be involved not only in physiological regeneration during the menstrual cycle, but also in the formation of endometrioid heterotopias. Two distinct types of stem cells have been identified in the endometrium: mesenchymal stromal cells (MSC) localized in the basal layer (eMSC) [16] and less studied epithelial stem/progenitor cells [17]. eMSC are phenotypically similar to MSC isolated from other tissues and capable of multipotent differentiation [18,19,20]. Since large quantities of eMSC are present in the menstrual blood [21], they can be brought to the peritoneal cavity during retrograde menstruations. It is assumed that eMSC can form the stroma of endometrioid heterotopias. Supposedly, the MSC of the endometrioid heterotopias (gMSC) can spread further by lymphatic and hematogenic routes and contribute to developing more endometrium-like foci by setting up their stroma, and probably even by producing their epithelial cells [22,23]. Interestingly, only eMSC, but not epithelial progenitor cells, can induce the growth of human endometrium-like tissue containing endometrial glands after transplantation into NOD-SCID mice [24].

It has been experimentally established that differences between eMSC and gMSC include a higher proliferative and migratory activity of the latter combined with an enhanced secretion of angiogenic factors [25]. These distinctions are probably due to the characteristics of the microenvironments of MSC in eutopic and ectopic endometrium (eutopic and ectopic niches) [26]. The ectopic niche probably includes immune cells as an essential element involved in regulating the progression of endometriosis either through direct contact with gMSC or by a paracrine mechanism.

The emergence and progression of endometriosis are accompanied by significant changes in the immune system and the formation of an inflammatory microenvironment in the ectopic niche. At the same time, the sensitivity of the ectopic endometrium to the effectors of cellular immunity changes significantly. Early studies in this field have demonstrated a strong inhibition of specific T cell-mediated cytotoxicity to autologous endometrial cells in endometriosis [27]. Much interest is drawn to the inability of natural killers (NK) to eliminate ectopic endometrium. NK failure to stop the progression of ectopic endometriosis may be due to the increased expression of inhibitory NK receptors, the presence of human leukocyte antigen G (HLA-G) at the surface of endometrial cells, or other mechanisms of suppression [28]. Thus, immunosuppression in the proinflammation environment is an essential factor in the pathogenesis of endometriosis. On the other hand, there is ample evidence that MSC can exert negative immunoregulation mediated by the pro-inflammatory factors [29].

The data presented above substantiate further comparative assessment of the immunoregulatory properties of eMSC and gMSC. In this study, we evaluated the ability of MSC isolated from eutopic and ectopic endometrium of the endometriosis patients to affect in vitro the critical mechanisms of adaptive immunity, namely the differentiation of monocytes into dendritic cells (DCs), the maturation of DCs, and the antigen-induced T-lymphocyte proliferation.

## 2. Experimental Section

### 2.1. Endometrial Tissue Samples and Mesenchymal Stromal Cell Cultures

The primary cultures of endometrial stromal cells were derived from the biopsies taken in the National Medical Research Center of Obstetrics, Gynecology, and Perinatology from the patients with histologically confirmed extragenital endometriosis of the abdominal cavity. The study was approved and supervised by the Institutional Ethics Committee (Approval Protocol No. 11, 7 December 2017). Written informed consent was obtained from all patients. Biological samples were handled using anonymous codes in accordance with the Federal Law on Personal Data (No. 152-FZ, 27 July 2006). Pipelle biopsy samples of eutopic endometrium and samples of ectopic endometrium retrieved during surgical removal of the endometrioid heterotopias of the gut were used in this study. The tissues were put into cold PBS (PanEco, Moscow, Russia), thoroughly minced using a scalpel, transferred to Hanks’s solution (PanEco, Moscow, Russia) containing 100 units/mL of collagenase type IV (Sigma-Aldrich, St. Louis, MC, USA), and incubated for 2 h at 37 °C. The generated suspension was filtered through the 40-micron pore diameter filter (Corning Life Sciences, Tewksbury, MA, USA) and twice washed by centrifugation at 300× *g*. The final pellet containing the endometrial cells was transferred into T75 cell culture flasks and maintained at standard culture conditions (37 °C, 5% CO_2_) in the DMEM/F-12 GlutaMAX medium supplemented with 10% fetal bovine serum (FBS) and the antibiotic/antimycotic mix (all components from Thermo Fisher Scientific, MA, Waltham, USA). The cells were allowed to adhere for two days, and non-adherent cells were removed by replacing the medium. Upon reaching 70–90% confluence, adherent cells were harvested by trypsinization and subcultured at 1:3 ratio in T75 cell culture flasks or 24-well plates. The cell cultures were cryopreserved at the 2nd passage in liquid nitrogen and thawed to be used as needed. The cells were counted using the Countess II (Thermo Fisher Scientific, Waltham, MA, USA) cell counter. Cell cultures were observed and photographed employing the Axiovert 40 CFL (Carl Zeiss, Oberkochen, Germany) inverted microscope and the Nikon D5000 digital camera (Nikon, Tokyo, Japan).

### 2.2. Dendritic Cell Differentiation and Maturation in the MSC Co-Culture

Mononuclear cells were isolated from the peripheral blood of healthy donors by Ficoll (1.077 g/mL) (PanEco, Moscow, Russia) density gradient centrifugation. Monocytes were obtained by immunomagnetic separation using the Dynabeads™ Untouched™ Human Monocytes Kit (Thermo Fisher Scientific, Waltham, MA, USA), transferred to the 24 well cell culture plates (Corning Life Sciences, Tewksbury, MA, USA) containing pre-plated MSC (2 × 10^5^ cells per well), and maintained in the RPMI-1640 GlutaMAX medium (Thermo Fisher Scientific, Waltham, MA, USA) supplemented with 10% heat-inactivated FBS (Thermo Fisher Scientific, Waltham, MA, USA), 50 µg/mL gentamycin (PanEco, Moscow, Russia), 5 мM HEPES buffer (STEMCELL Technologies Inc., Vancouver, BC, Canada), and 100 µg/mL sodium pyruvate (Sigma-Aldrich, St. Louis, MC, USA) at normal culture conditions (37 °C, 5% CO_2_). Monocytes-to-DCs differentiation was induced by the addition of granulocyte-macrophage colony-stimulating factor (GM-CSF) and interleukine 4 (IL4) (both from ProSpec, Ness-Ziona, Israel) at final concentrations of 80 and 50 ng/mL, respectively. Differentiation efficacy was evaluated on the fourth co-culture day by the onset of CD1a expression and loss of CD14 expression. DCs maturation was stimulated by adding 1 µg/mL lipopolysaccharide (LPS) (Sigma-Aldrich, St. Louis, MC, USA). The degree of the DCs maturity was assessed 48 h later by the appearance of the CD83 co-stimulation molecule combined with the enhanced expression of HLA-DR. Wells with desquamated MSC monolayer were excluded from the subsequent analysis.

### 2.3. T-Lymphocyte Proliferation in the MSC Co-Culture

Lymphocytes were separated from monocytes by the incubation of blood mononuclear cells for 2 h at 37 °C and 5% CO_2_ in the RPMI-1640 GlutaMAX medium supplemented with 10% heat-inactivated FBS, 50 µg/mL gentamycin, 5 мM HEPES, and 100 µg/mL sodium pyruvate. Non-adhesive cells (lymphocytes) were collected, stained with 5 µM carboxyfluorescein diacetate succinimidyl ester (CFSE) (Sigma-Aldrich, St. Louis, MC, USA) following the standard protocol [30], transferred to 24-well plates containing pre-plated MSC, and co-cultured with them for 4 days. T-lymphocyte proliferation was induced by the addition of the Dynabeads™ Human T-Activator CD3/CD28 for T Cell Expansion and Activation (Thermo Fisher Scientific, Waltham, MA, USA) that simulates antigens and then evaluated by the intensity of the CFSE fluorescence.

### 2.4. Flow Cytometry

For flow cytometric analysis, the cells were resuspended in PBS, supplemented with 2% heat-inactivated FBS and 0.05% sodium azide, and stained for 30 min at 4 °C with monoclonal antibodies against cell surface markers (BD Bioscience, Franklin Lakes, NJ, USA) conjugated with fluorescein isothiocyanate (FITC), phycoerythrin (PE), allophycocyanin (APC), or PE-Cy™5 tandem fluorochrome. Stained cells were washed with PBS twice, fixed with Cytofix (BD Bioscience, Franklin Lakes, NJ, USA), and analyzed using the FACSAria III flow cytometer/cell sorter (BD Bioscience, Franklin Lakes, NJ, USA). Non-specific isotypic antibodies conjugated with the same fluorochrome served as negative controls. FACSDiva 7 and FlowJo V10 software (BD Bioscience, Franklin Lakes, NJ, USA) were used to process the flow cytometric data.

To assess the proliferation of lymphocytes during co-cultivation with MSC, all non-adherent cells from every single well were suspended in 2 mL of the medium. Then, 1 mL of each suspension was used to compare cell counts in individual wells at identical instrument parameters (sample pressure—5, sheath pressure—70 psi, t = 120 s) in the absence of beads. The rest of each suspension was used for the CFSE distribution analysis. In both cases, the lymphocytes were gated according to their side and forward scatter parameters.

### 2.5. Multiplex Analysis of the Cytokine Secretion

The profiling of the cytokines secreted by MSC was carried out by the multiplex immunoenzyme analysis. The media conditioned by MSC monolayer cultures were collected at the third passage, centrifuged, and the supernatants were frozen and kept at −80 °C. Cytokine concentrations were measured in undiluted supernatants and at 1:10 dilution using the Bio-Plex Pro Human Chemokine Panel 40-plex Assay and the laser immunoanalyzer Bio-Plex 200 equipped with Bio-Plex Manager 6.0 Properties software (both assay and analyzer from Bio-Rad Laboratories, Hercules, CA, USA) according to the manufacturer’s protocol.

### 2.6. Statistical Analysis

All measurements were performed in at least three replicates. Statistical differences between groups were calculated by Student’s *t*-test or non-parametric Mann–Whitney U-test; *p*-values of less than 0.05 were regarded as significant.

## 3. Results

### 3.1. Isolation of Cell Cultures from Eutopic and Ectopic Endometrium

In this study, cultures of MSC originating from the biopsies of eutopic (eMSC) and ectopic (gMSC) endometrium of six patients with extragenital endometriosis were established and characterized. All cultures actively proliferated and consisted of cells with fibroblast-like morphology typical for MSC (Figure 1). At the same time, there were slight differences in cell shape, most evident in cultures from patient #5, where eMSC were more compact and less spread out than gMSC. It was probably due either to the different microenvironment in vivo or to distinct lineage descent of eMSC and gMSC. The established cell cultures were analyzed at 3–6 passages to compare their molecular markers and immunological properties.

### 3.2. Expression of Cell Surface Molecular Markers by Cultures Isolated from Eutopic or Ectopic Endometrium

All eMSC and gMSC cultures were subjected to flow cytometry to analyze their phenotype and immunological features. The results of the comparative cytometric analysis of eMSC and gMSC obtained from one of the patients are presented in Figure 2. The cultures initiated from eutopic and ectopic endometrium expressed MSC markers CD73, CD90, and CD105, while hematopoietic cell markers CD34, CD45, CD11b, CD19, and epithelial cell markers CD24 and EpCAM were absent. Therefore, the two studied types of cell cultures consisted of cells of stromal descent with CD-phenotype usual for MSC [25].

Remarkably, both eMSC and gMSC actively expressed the CD44 hyaluronic acid receptor. High levels of this cell surface receptor are known to be associated with the stem cell phenotype. CD44 is involved in MSC migration and homing [31]. It is also commonly referred to as a cancer stem cell marker [32].

Elsewhere, we reported that the cell adhesion molecule CD54 (ICAM-1) is a marker distinguishing human MSC from embryonic and adult skin fibroblasts [33]. ICAM-1 is the ligand of β2–integrins expressed at the surface of blood cells. It plays an essential role in the interactions between tissues and immune cells, particularly in inflammatory reactions [34]. Both eMSC and gMSC isolated from endometriosis patients had high levels of the expression of this marker. At the same time, the two cell types only weakly expressed Class I Major Histocompatibility Complex (HLA-ABC) and did not express Class II Major Histocompatibility Complex (HLA-DR) molecules or co-stimulation molecules CD80, CD83, and CD86 involved in the induction of the immune response. Both eMSC and gMSC showed the same moderate level of expression of the tumor necrosis factor receptor CD95 involved in inducing apoptosis by the immune cells.

In summary, we found no reliable differences in the expression of the studied surface molecules by the cells of cultures derived from the eutopic and ectopic endometrium. Similar expression profiles were found in cultures derived from all patients.

### 3.3. Influence of the MSC Cultures Isolated from the Eutopic or Ectopic Endometrium on the Differentiation and Maturation of DCs In Vitro

DCs play an essential role in adaptive immunity since they can process antigens and present them to the lymphocytes, thus initiating the primary immune response [35]. DCs are derived by the differentiation of monocytes or CD34-positive hematopoietic stem cells. Depending on the degree and way of further maturation, they can stimulate or downregulate immune reactions. Assuming that the interactions of MSC from eutopic and ectopic endometrium with DCs can constitute an essential stage of the pathogenesis of endometriosis, we studied these processes using an in vitro model.

To assess the possible influence of the MSC cultures on the differentiation of DCs from monocytes, allogeneic monocytes from the peripheral blood of healthy donors were co-cultured with eMSC or gMSC. DCs differentiation was induced by adding recombinant GM-CSF and IL4 to the culture. The differentiation efficiency was assessed by flow cytometry using the monocyte marker CD14 and the DCs marker CD1a.

Co-culture experiments demonstrated that just one of the six eMSC cultures (eMSC-3) caused weak suppression of DCs differentiation (Figure 3A). The other five cultures, including eMSC-5 shown in Figure 3B, did not affect this process. Unlike eMSC, all gMSC cultures effectively blocked DCs differentiation. Pooled data from co-culture experiments (Figure 3C) show no significant difference between the efficacy of DCs differentiation in monoculture and eMSC co-culture (*p* = 0.350). At the same time, the strong suppressive effect of stromal cells of endometrioid heterotopias on the DCs differentiation is beyond doubt (*p* = 0.005).

To assess the effect of the studied cultures on the maturation of DCs, MSC were co-cultured with immature DCs obtained from monocytes due to their induced differentiation. The maturation of DCs in the co-culture was stimulated by adding LPS to the culture medium. The maturation efficiency was assessed by the appearance of the co-stimulating molecule CD83 on the DCs surface and the increase of the HLA-DR expression. In our experiments, neither eMSC nor gMSC cultures affected DCs maturation (Figure 4A) even at a 1:1 ratio. Only gMSC-3 culture made an exception (Figure 4C), since it reliably (*p* = 0.032, n = 3), though not strongly, suppressed this process.

Thus, despite the similarity of phenotypic characteristics, stromal cell cultures generated from eutopic and ectopic endometrium demonstrated different abilities to suppress the differentiation of DCs. Somewhat atypical immunological properties of both cultures obtained from patient #3 could be associated with the specific features of the patient’s pathology, since the characteristics of the endometrium of patients with endometriosis, including its immunological properties, can vary significantly [27].

### 3.4. Influence of MSC Cultures Derived from Eutopic or Ectopic Endometrium on T-Cell Proliferation In Vitro

T-lymphocytes play a leading role in the effector and regulatory mechanisms of cellular immunity and are most likely involved in the pathogenesis of endometriosis. Therefore, the MSC cultures were tested for their ability to stimulate or suppress the proliferation of T-lymphocytes in vitro. MSC isolated from the eutopic or ectopic endometrium were co-cultured with allogeneic lymphocytes isolated from the peripheral blood of healthy donors. Some of them were stimulated to proliferate with magnetic particles carrying the antibodies against CD3 and CD28 that simulate contact with antigen. The proliferative activity was determined by analyzing the distribution of the CFSE dye among the cells and by direct counting of the lymphocyte’s number after the co-cultivation.

The cells of all cultures generated from the eutopic endometrium had virtually no effect on the induced lymphocyte proliferation. In contrast, cells derived from the ectopic endometrium suppressed the lymphocyte proliferation even at an initial ratio of 1:10 (Figure 5). Significant differences in the suppressive capacity of the two types of MSC were revealed both by the analysis of the CFSE distribution (*p* < 0.001) and by calculation of the number of lymphocytes after co-cultivation (*p* = 0.001). As seen on the histograms (Figure 5A), despite the suppressive effect of gMSC, some of the stimulated lymphocytes still pass the first division. The exclusion of the first division from the analysis clearly shows that gMSC completely suppress lymphocyte proliferation (Figure 5B,C). This conclusion is fully supported by the lymphocyte’s number calculation data (Figure 5D).

### 3.5. Comparative Profile of Cytokine Secretion by Cell Cultures of Eutopic and Ectopic Endometrium

Modulating the immunity by MSC isolated from various organs and tissues is well established [29,36]. It is commonly accepted that MSC induce immunomodulation by secreting several cytokines and other biologically active agents [37]. Based on this, we compared culture media conditioned by the eutopic or ectopic endometrium cultures using a multiplex immunoassay system that allows assessing 40 cytokines per run.

We found that the stromal cells of the endometrioid heterotopias produce much more inflammation-associated cytokines. For instance, the mean IL6 concentration in the media conditioned by the gMSC was approximately 180 times higher than in the eMSC-conditioned media (Table 1). Interestingly, it has been reported that MSC-secreted IL6 not only stimulates inflammation but also participates in the suppression of DCs differentiation from the bone marrow progenitors [38]. Another pro-inflammatory factor, IFN-γ, was also elevated in the media conditioned by the gMSC cultures. IFN-γ is known to combine pro-inflammatory features with the ability to initiate the immunosuppressive MSC phenotype [39]. In addition, the enhanced secretion of another important pro-inflammatory cytokine, IL1b by the gMSC, though statistically non-significant, should also be taken into account. Besides, gMSC cultures-conditioned media contain much higher concentrations of the inflammation-associated chemokines. The CCL2, CXCL6, CCL7, CCL8, CXCL9, and CCL17 chemokines can attract different immune cells, including monocytes and lymphocytes, activating inflammation [40]. On the other hand, we did not find substantial differences between eMSC and gMSC in the secretion of another important pro-inflammatory cytokine, TNF-α. The concentration of IL10, usually associated with negative immunoregulation, also did not differ in the two types of cell cultures (Appendix A).

Thus, the analysis of the secretomes demonstrated that MSC isolated from the endometrioid heterotopias have a pro-inflammatory phenotype and likely contribute to the pro-inflammatory microenvironment within the endometrioid lesion, supporting the progression of the extragenital endometriosis.

## 4. Discussion

It is believed that the central role in the formation of the immunosuppressive and inflammatory background in endometrioid lesions belongs to the immune cells, such as activated macrophages secreting the pro-inflammatory factors [41], myeloid-derived suppressor cells (MDSCs) [42], and regulatory T-cells (Tregs) [43]. In this study, we showed that the main stromal component of the endometrioid heterotopias, gMSC, can by itself almost completely suppress DCs and T-lymphocytes that are deeply involved in critical processes of adaptive immunity. Moreover, gMSC actively secrete cytokines capable of forming an inflammatory zone and recruiting immune cells into it. Thus, it is possible that just gMSC trigger the mechanism predetermining the special immunological conditions within the endometrioid heterotopias that may be a way of escaping immunological control. This assumption is supported by a report that the depletion of DCs in a mouse in vivo model reduces the number of activated T-lymphocytes and accelerates the growth of the endometriosis foci [44].

The ability of MSC of varying origin to suppress DCs differentiation [45,46] and lymphocyte proliferation [47] has been well established. Several mechanisms by which MSCs suppress immune cells are described in the literature. They include direct cell-to-cell contacts and paracrine interactions. IFN-γ alone [39,48] or in combination with TNF-α and IL1β [49] is commonly regarded as a signal switching on the immunosuppressive state of MSC. In our experiments, we demonstrated the enhanced secretion of IFN-γ from gMSC (Table 1) as well as the moderate secretion of IL1β and TNF-α from both eMSC and gMSC (Appendix A). In this regard, it is possible that gMSC maintain the immunosuppressive phenotype by autocrine regulation.

In both the DCs differentiation [50] and lymphocyte proliferation [51] experiments, MSC can directly cause irreversible cell cycle arrest at the G0/G1 phase. Another mechanism is related to the secretion of IL6 [38]. IL6 up-regulates M-CSF receptors’ expression on monocytes, making them sensitive to the autocrine M-CSF, and shifting monocyte differentiation towards the macrophage [52]. Since we observed a significant increase in IL6 secretion by gMSC compared to eMSC, the involvement of this mechanism in negative immunoregulation demonstrated in our experiments is highly probable. Indeed, the number of both M1 and M2 macrophages was significantly higher in ovarian endometriomas than in functional ovarian cysts [53]. Furthermore, the switch of the monocyte differentiation from DCs to macrophages has also been reported in the peritoneal fluid of endometriosis patients [54]. In this regard, it is worth noting that in endometriosis, the level of IL6 in the peritoneal fluid is usually elevated compared to healthy women and shows a positive correlation with the size and quantity of the endometrioid lesions [55].

It has been hypothesized that MSC can be polarized towards either a pro-inflammatory or an immunosuppressive phenotype depending on the prevailing TLR signals [56]. MSC1 display the pro-inflammatory cytokine secretion profile and stimulate lymphocyte activation in co-culture. In contrast, MSC2 have an immunosuppressive profile and suppress the activation of lymphocytes. However, in our study, gMSC had a pro-inflammatory cytokine profile and induced pronounced immunosuppressive effects in vitro. The apparent contradiction between the ability to suppress immune processes and the pro-inflammatory status may be deceptive. Indeed, the enhanced expression of the CLTA-4 immunosuppressive molecule characteristic of the anergic state of lymphocytes is associated with the maintenance of chronic inflammation in endometriosis [57]. Moreover, the effects of some pro-inflammatory cytokines, including IL6, depend on their concentration, the state of target cells, and other factors. It has been shown that IL6-secreting MSC can sustain or suppress lymphocyte proliferation in a co-culture depending on the lymphocyte/MSC ratio [58]. Possibly, the ability to turn off certain adaptive mechanisms, specifically in an inflammatory environment, is a feature of MSC associated with their role in immune system regulation.

A certain similarity between endometrioid heterotopies and malignant tumors is obvious. In this regard, applying the theory of cancer stem cells (CSCs) to the pathogenesis of endometriosis may be helpful. According to modern views, CSCs exhibit plasticity and can acquire or lose their stemness depending on the microenvironment [59]. These changes are closely related to the epigenetic profile of the CSCs. There is accumulating evidence that epigenetic changes may also play a pivotal role in forming endometrioid heterotopias [60]. If so, the stemness of endometrioid cells may be maintained by pro-inflammatory cytokines and other factors of the ectopic niche. Following this concept, it can also be assumed that the population of endometrioid cells is hierarchical and consists of cells of varying degrees of differentiation. In this case, the least differentiated cells (endometrioid stem cells) may be responsible for relapses and the dissemination of extragenital endometriosis.

The origin of the ectopic stromal cells can be directly related to their role in immunoregulation. Are gMSC descendants of the eutopic endometrium cells? Are they the result of differentiation of circulating endometrioid stem cells, or are they former eMSC transformed by the ectopic microenvironment? Alternatively, the ectopic endometrium stroma could be formed by attracting MSC-like cells, which were initially immunosuppressive. Recently, these questions were partly answered by applying target-gene or whole-exome sequencing and droplet digital PCR methods. No mutations shared by the epithelium and stroma of the endometrioid lesions were revealed [61,62], arguing for their independent origin. Moreover, the analysis of the selection-neutral passenger mutations demonstrated the clonality of the epithelial but not stromal component of the ectopic endometrium [59]. We hypothesize that the special immunological features of the endometrioid heterotopias may result from the recruitment of resident MSC or fibroblasts or bone marrow MSC known to display stable pronounced immunomodulation activity mediated by IL6 [29,36,63].

## 5. Conclusions

Despite similar morphology and molecular phenotype, MSC isolated from the stromal part of the eutopic endometrium (eMSC) and the endometrioid heterotopias (gMSC) have different immunological properties. Unlike eMSC, gMSC are able to almost completely suppress the differentiation of DCs from monocytes and the T-lymphocyte proliferation in vitro. At the same time, MSC from ectopic endometrioid lesions display a pronounced pro-inflammatory cytokine secretion profile. The immunological characteristics of gMSC may strongly contribute to the immunological properties of the endometrioid heterotopias and probably to the disease progression.

## Figures and Tables

**Figure 1 biomedicines-09-01286-f001:**
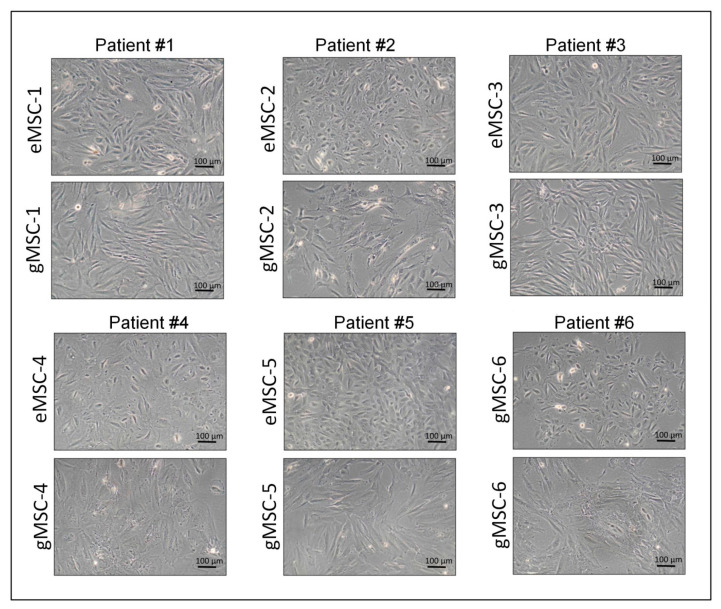
Cell cultures initiated from eutopic endometrium (eMSC) and endometrioid heterotopias (gMSC) of the patients with extragenital endometriosis (passage 3). Phase contrast. Scale bar: 100 microns.

**Figure 2 biomedicines-09-01286-f002:**
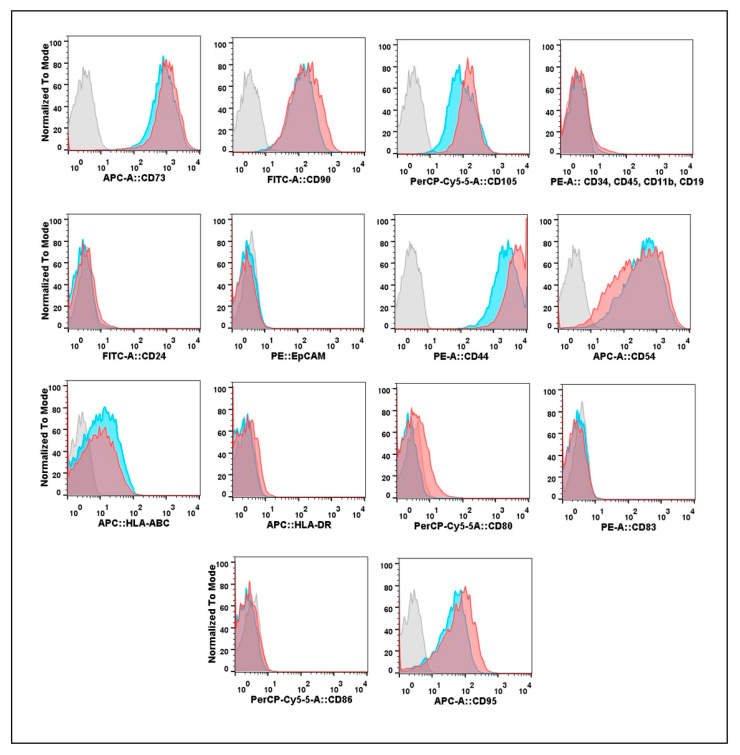
Expression of surface molecular markers on the mesenchymal stromal cell (MSC) cultures isolated from eutopic endometrium (eMSC) and endometrioid heterotopias (gMSC). Cell suspensions were treated with fluorochrome-conjugated monoclonal antibodies (MoAbs) and analyzed by flow cytometry *X*-axis—fluorescence intensity; *Y*-axis—normalized to mode percentage of cells. Blue curves—eMSC-5; red curves—gMSC-5; gray curves—gMSC-5 isotypic control. Since there are no significant differences between gMSC and eMSC isotypic controls, the eMSCs isotypic control histograms are not shown.

**Figure 3 biomedicines-09-01286-f003:**
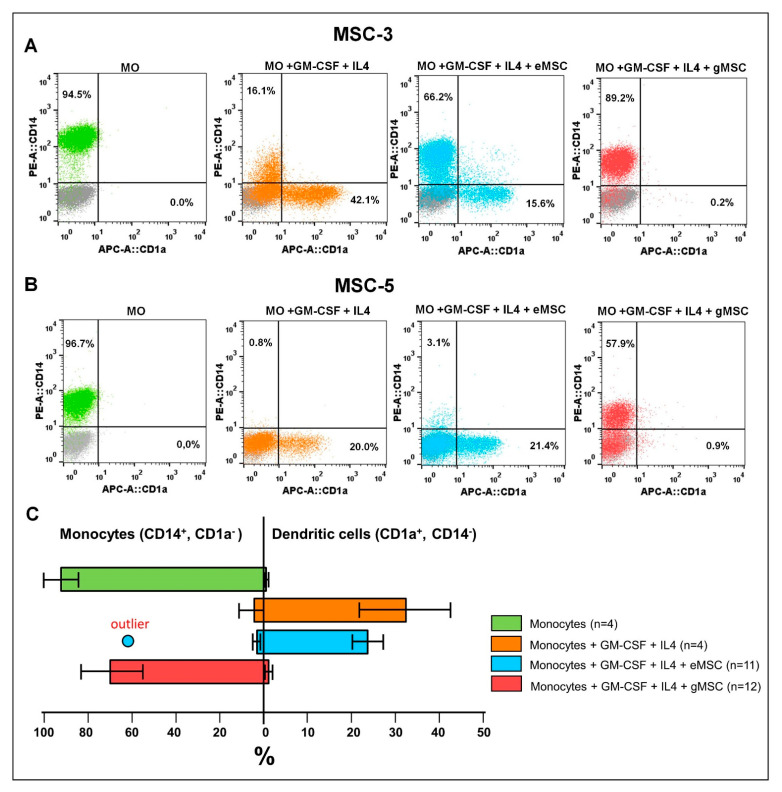
Effect of stromal cells from eutopic and ectopic endometrium upon the differentiation of dendritic cells (DCs) in vitro. Monocytes (MO) were co-cultured with stromal cells of eutopic (eMSC) or ectopic (gMSC) endometrium in the presence of GM-CSF and IL4 (monocytes/MSC ratio 5:1). After 4 days in co-culture, the cells were double-stained with fluorescent MoAbs against CD14 and CD1a and analyzed by flow cytometry to calculate the percentage of monocytes and DCs. (**A**,**B**) Results for co-cultures with MSC-3 and MSC-5 (typical dot plots). Gray dots denote isotype controls. (**C**) Processed data combining all measurements of six pairs of MSC co-cultures presenting the percentage of cells in the subpopulations and SD.

**Figure 4 biomedicines-09-01286-f004:**
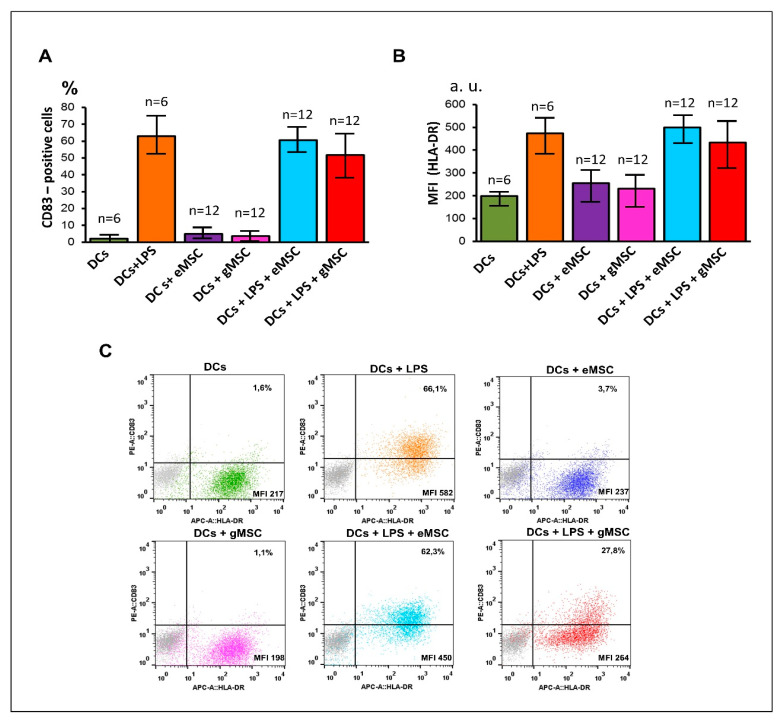
Effects of stromal cells from eutopic or ectopic endometrium upon the maturation of dendritic cells (DC) in vitro. Immature DCs were co-cultured with stromal cells of eutopic (eMSC) or ectopic (gMSC) endometrium with or without LPS (DCs/MSC ratio 1:1). After 2 days, co-culture cells were double stained with fluorescent MoAbs against CD83 and HLA-DR and analyzed by flow cytometry to evaluate the percentage of DCs expressing the maturation marker CD83 (**A**) and the level of HLA-DR expression (**B**). Processed data combining all measurements of five pairs of MSC co-cultures are presented (**A**,**B**). Whiskers show the SD. MFI—median fluorescence intensity measured in arbitrary units (a.u.). (**C)** The results of MSC-3 co-cultures (typical dot plots); gray dots denote isotype controls.

**Figure 5 biomedicines-09-01286-f005:**
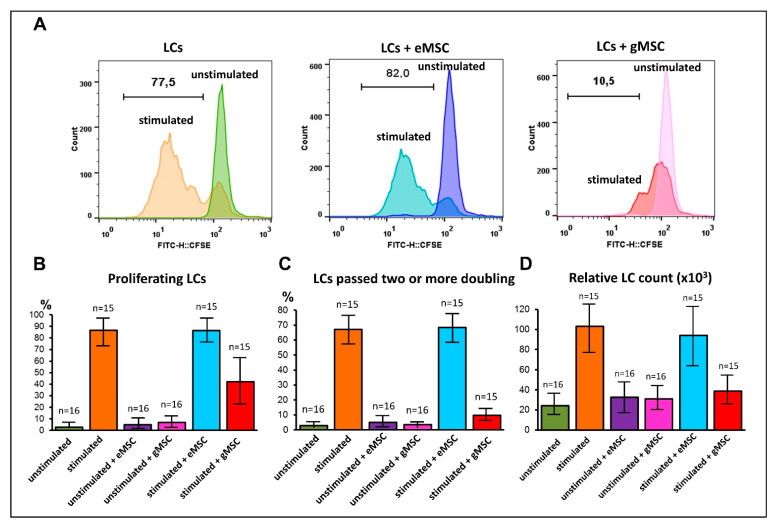
Effects of stromal cells from eutopic or ectopic endometrium upon the proliferation of T-lymphocytes in vitro. Lymphocytes (LCs) were stained with CFSE, stimulated to proliferate with “Dynabeads”, and co-cultured with stromal cells of eutopic (eMSC) or ectopic (gMSC) endometrium (LCs/MSC ratio 10:1). After 4 days in co-culture, the cells were analyzed by flow cytometry to evaluate CFSE distribution and relative LSs counts. (**A**) distribution of CFSE in MSC-6 co-cultures (typical histograms). (**B**–**D**) Processed data combining all the measurements of six pairs of the MSC co-cultures. (**B**) The percentage of proliferating LCs. (**C**) The percentage of proliferating LCs having passed two or more doubling. (**D**) Relative LCs counts. Whiskers represent SD.

**Table 1 biomedicines-09-01286-t001:** Comparison of cytokine concentration in the culture media conditioned by eMSC or gMSC. Cytokine concentrations differing 2.5 times or more in at least five pairs of cell cultures are presented.

Cytokine	Cell Cultures						Fold Change	*p*-Value
	eMSC			gMSC				
	Mean (pg/mL)	SD	SEM	Mean (pg/mL)	SD	SEM		
IL1b	0.77	1.38	0.56	8.02	16.14	6.59	10.4	0.18
IL6	222.46	191.36	78.12	40,067.58	42,742.52	17,449.56	180.1	0.002
IFN-γ	3.45	4.32	1.76	20.26	11.07	4.52	5.9	0.004
CTACK/CCL27	0.59	0.55	0.23	2.67	1.71	0.70	4.5	0.018
GCP-2/CXCL6	2.23	2.96	1.21	180.61	208.40	85.08	81.0	0.002
MCP-1/CCL2	157.60	226.60	92.51	2647.84	2037.01	831.60	16.8	0.002
MCP-2/CCL8	0.73	1.07	0.44	16.98	2.42	11.19	23.2	0.026
MCP-3/CCL7	13.49	10.82	4.41	166.18	118.37	48.32	12.3	0.002
MIG/CXCL9	4.90	3.72	1.52	22.33	5.43	2.21	4.5	<0.001
TARC/CCL17	3.44	4.87	1.99	19.79	7.74	3.16	5.7	0.001

## Data Availability

The data presented in this study are available on request from the corresponding author.

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
