# Peer review of "Mesenchymal Stromal Cells Isolated from Ectopic but Not Eutopic Endometrium Display Pronounced Immunomodulatory Activity In Vitro"

_biomedicines, 2021, doi:10.3390/biomedicines9101286_

Round 1
Reviewer 1 Report
The manuscript by A.Yu Lupatov, et al. addresses an important issue about progression of endometriosis and its association with the realignments of the immune system. Endometriosis, a common condition that affects up to 10% of women of childbearing age is caused by the anomalous accumulation of endometrial cells outside the uterus and often seriously compromises the quality of life. Recent studies have shown that stem cells that are found in normal endometrium and are important for physiological regeneration during the menstrual cycle could be a source for the formation of endometrioid heterotopias. Several distinct types of stem cells have been identified in endometrium including mesenchymal stromal cells (eMSC) that are localized in the basal layer. It is believed that eMSC are able form the stroma of endometrioid heterotopias, and the cells from endometrioid heterotopias (gMSC) can spread further and contribute to the development of more endometrium-like foci. The progression of endometriosis is associated with notable realignments of the immune system and local inflammatory.
In the presented study authors evaluated the ability of mesenchymal stromal cells originated from eutopic or ectopic endometrium to interfere with adaptive immunity mechanisms. They have produced primary cultures from biopsies of normal endometrium and endometrioid heterotopias of patients with extragenital endometriosis. They specifically studied the differentiation of monocytes into dendritic cells (DCs), the maturation of DCs, and the mitogen-induced lymphocyte proliferation and have shown that cells originated from endometrioid heterotopias supress dendritic cell differentiation from monocytes and from lymphocyte proliferation in allogeneic co-cultures. Cells originated form eutopic endometrium were unable to suppress this differentiation. Authors have analysed secretomes and identified increased level of several markers of inflammation that have been induced by cells from endometrioid heterotopias. Thereby authors produced the direct evidence supporting the suggestion that endometrioid heterotopias could exert pro-inflammatory and immunosuppressive activities.
This is a thoroughly and well executed study, its conclusions are clear and supported by the data. While the experiments outlined are interesting and provide value to the field, the manuscript requires some improvement prior to publication:
- There are published data indicating that MSC isolated from a number of tissues (bone marrow, placenta, etc.) suppress not only DC differentiation from monocytes, but DC maturation as well (references given below). In the reviewed manuscript MSC-like stromal cells isolated from ectopic endometrioid lesions suppressed just DC differentiation, but not maturation. How do the authors explain this?
Liu Y, Yin Z, Zhang R, Yan K, Chen L, Chen F, Huang W, Lv B, Sun C, Jiang X. MSCs inhibit bone marrow-derived DC maturation and function through the release of TSG-6. Biochem Biophys Res Commun. 2014 Aug 8;450(4):1409-15. doi: 10.1016/j.bbrc.2014.07.001.
Abomaray, F.M., Al Jumah, M.A., Kalionis, B. et al. Human Chorionic Villous Mesenchymal Stem Cells Modify the Functions of Human Dendritic Cells, and Induce an Anti-Inflammatory Phenotype in CD1+ Dendritic Cells. Stem Cell Rev and Rep 11, 423–441 (2015). https://doi.org/10.1007/s12015-014-9562-8
Liu WH, Liu JJ, Wu J, Zhang LL, Liu F, Yin L, Zhang MM, Yu B. Novel mechanism of inhibition of dendritic cells maturation by mesenchymal stem cells via interleukin-10 and the JAK1/STAT3 signaling pathway. PLoS One. 2013;8(1):e55487. doi: 10.1371/journal.pone.0055487.
- “Cytophenotype” is not a commonly used term, “phenotype” would be more appropriate.
- In the "Experimental Section" the authors indicate that the stimulation of T-lymphocyte proliferation was carried out using the Dynabeads™ Human T-Activator CD3/CD28.Strictly speaking, CD3/CD28 activation with specific antibodies is not “mitogen-induced” T-lymphocyte proliferation, since the antibodies in question are not mitogens like, for example, Concanavalin A. Therefore, it will be better to avoid using “mitogen-induced” to denote CD3/CD28 activation with specific antibodies.
- The sentence "The generated suspension was filtered through the 40 mm pore diameter filter" clearly contains a typo regarding the pore size (40 mm = 4 cm).
- Typically, heat-inactivated serum is used for culturing immune cells. The authorsdonot indicate this.
- The authors use "pyruvate" instead of "Sodium pyruvate", which is incorrect. They also give varying Sodium pyruvate concentrations in the culture media: 100 g/ml in section 2.2 versus 100 mg/ml in section 2.3. Where is thetypo?
- The concentration of CFSE used for staining lymphocytes was not specified.
- What software did the authors use to process the flow cytometric data?
- At the end of the first paragraph of section 3.2, a reference to Figure 2 should be included.
- In Figure 2, there is just one isotypic control for each histogram. Are the peaks exactly the same for the two cell types (eMSCand gMSC)?
- In the caption to Figure 2, the authors wrote "Y-axis - a percentage of cells in a population". Usually, the Y-axis on the flow cytometric histograms does not reflect the percentage of cells. Is there a mistake in the text and Y-axis shows cell counts?
- Figure 4 needs to be moved one paragraph below.
- On a number of occasions the authors use a strange designation of micrograms (presumably): “ug”.
- How was the choice of the DC/MSC ratio made in the co-cultivation experiments? Why were different cell ratios used in the differentiation (5:1) and maturation (1:1) of DCs experiments?
Author Response
Dear reviewer,
Thank you very much for the detailed revision of our manuscript entitled “Mesenchymal stromal cells isolated from ectopic but not eutopic endometrium display pronounced immunomodulatory activity in vitro” (manuscript ID: biomedicines-1383507- minor revisions). Your suggestions were very helpful for improving our manuscript. We have made revisions according to your comments, as described in the authors' response. Please see the attachment.

Reviewer 2 Report
I read with great interest the manuscript, which falls within the aim of this Journal. In my honest opinion, the topic is interesting enough to attract the readers’ attention. Nevertheless, authors should clarify some points and improve the discussion, as suggested below.
Authors should consider the following recommendations:
- Manuscript should be further revised in order to correct some typos and improve style.
- Accumulating evidence suggests that immune cells, adhesion molecules, extracellular matrix metalloproteinase and pro-inflammatory cytokines activate/alter peritoneal microenvironment, creating the conditions for differentiation, adhesion, proliferation and survival of ectopic endometrial cells. I would discuss these points in the light of new theories about the pathogenesis of endometriosis, referring to: PMID: 31663401; PMID: 28100109.
Author Response
Dear reviewer,
Thank you very much for reading our manuscript entitled “Mesenchymal stromal cells isolated from ectopic but not eutopic endometrium display pronounced immunomodulatory activity in vitro” (manuscript ID: biomedicines-1383507- minor revisions) and very valuable suggestions. We have made revisions according to your comments, as described in the authors' response. Please see the attachment.
